# A pitfall for machine learning methods aiming to predict across cell types

Jacob Schreiber[1], Ritambhara Singh[2,3], Jeffrey Bilmes[1,4] and William Stafford Noble[1,2*]

*Correspondence:
william-noble@uw.edu
[1]Paul G. Allen School of Computer
Science & Engineering, University of
Washington, Seattle, USA
[2]Department of Genome Science,
University of Washington, Seattle,
USA
Full list of author information is
available at the end of the article

**Abstract**
Machine learning models that predict genomic activity are most useful when they make accurate predictions across cell types. Here, we show that when the training and test sets contain the same genomic loci, the resulting model may falsely appear to perform well by effectively memorizing the average activity associated with each locus across the training cell types. We demonstrate this phenomenon in the context of predicting gene expression and chromatin domain boundaries, and we suggest methods to diagnose and avoid the pitfall. We anticipate that, as more data becomes available, future projects will increasingly risk suffering from this issue.

**Keywords:**  Machine learning, Epigenomics, Genomics

Machine learning has been applied to a wide variety of genomic prediction problems, such as predicting transcription factor binding, identifying active *cis*-regulatory elements, constructing gene regulatory networks, and predicting the effects of single nucleotide polymorphisms. The inputs to these models typically include some combination of nucleotide sequence and signals from epigenomics assays.

Given such data, the most common approach to evaluating predictive models is a "cross-chromosomal" strategy, which involves training a separate model for each cell type and partitioning genomic loci into some number of folds for cross-validation (Fig. 1a). Typically, the genomic loci are split by chromosome. This strategy has been employed for models that predict gene expression [1–3], elements of chromatin architecture [4, 5], transcription factor binding [6, 7], and *cis*-regulatory elements [8–13]. Although the cross-chromosomal approach measures how well the model generalizes to new genomic loci, it does not measure how well the model generalizes to new cell types. As such, the cross-chromosomal approach is typically used when the primary goal is to obtain biological insights from the trained model.

An alternative, "cross-cell type" validation approach can be used to measure how well a model generalizes to a new cell type. This approach involves training a model in one or more cell types and then evaluating it in one or more other cell types (Fig. 1b). Note that

**Fig. 1** The performance of neural network models of varying complexity in three predictive settings on two tasks. Schematic diagrams of **a** cross-chromosome, **b** cross-cell type, and **c** hybrid cross-cell type/cross-chromosomal model evaluation schemes. **d**–**f** The figure plots the average precision (AP) of a machine learning model predicting gene expression as a function of model complexity. Evaluation is performed via **d** cross-chromosome, **e** cross-cell type, and **f** a combination of cross-chromosome and cross-cell type validation. In each panel, each point represents the test set performance of a single trained model. **g**–**i** is the same as **d**–**f** but predicting TAD boundaries rather than gene expression

in this setting, although the evaluation is done across cell types, the model is still predicting within a single cell type—e.g., predicting gene expression in a given cell type from epigenomic measurements in that same cell type. Researchers have used this approach to identify *cis*-regulatory elements [14–16], predict regions of accessible chromatin [17, 18] and impute epigenomics assays that have not yet been experimentally performed [19, 20]. The cross-cell type strategy is typically adopted when the goal is to yield predictions in cell types for which experimental data is not yet available.

In this work, we point out a potential pitfall associated with cross-cell type validation, in which this evaluation strategy leads to overly optimistic assessment of the model's performance. In particular, we observed that models evaluated in a cross-cell type setting seem to perform better as the number of parameters in the model increases. To illustrate this phenomenon, we train a series of increasingly large neural networks to predict gene expression as measured by RNA-seq in the H1 cell line (E003), evaluating each model using the cross-chromosomal and the cross-cell type approaches. As input, each model receives a combination of nucleotide sequence and epigenomic signal from examples in the H1 cell line or 55 other cell lines, depending on evaluation setting (see Additional file 1). In every case, we evaluate model performance using the average precision score relative to a binary gene expression label ("high" versus "low" expression). In the cross-chromosome setting, the performance of the models remains fairly constant as the complexity of the learned model increases (green points in Fig. 1d). On the other hand, the cross-cell type results show a surprising trend: using more complex models appears to yield consistently better results, even as the models become very large indeed (up to 100 million parameters; Fig. 1e).

To see that this apparently good predictive performance is misleading, we perform a third type of validation, a hybrid "cross-chromosome/cross-cell type" approach in which the model is evaluated on loci and cell types that were not present in the training set

(Fig. 1c). This approach has previously been used to identify *cis*-regulatory elements [21, 22] and to predict CpG methylation [23]. We found that evaluating models using the hybrid approach eliminates the positive trend in model performance as a function of model complexity (Fig. 1f). Very similar trends are seen when we train neural networks to predict the locations of topologically associating domain (TAD) boundaries in the H1 cell line (Fig. 1g–i). Further, these results do not appear to be specific to deep neural networks, as gradient-boosted decision tree classifiers show similar trends as the number of trees increases (Additional file 2: Figure S1). Note that the random baseline, which is the expected average precision when the predictions are uniformly random values and represents a lower bound of performance, differs between the cross-cell type and other settings because we use a different chromosome for the test set (see Additional file 1 for details).

The following three observations suggest that the positive trend in Fig. 1e arises because more complex models effectively "memorize" the genomic location associated with expressed versus non-expressed genes. First, if we train a model using only the epigenomic signal, without including the nucleotide sequence as input, then the model performance no longer improves as a function of model complexity (orange points in Fig. 1e); conversely, providing only nucleotide sequence as input yields very good performance across many cell types (blue points in Fig. 1e). Second, even when we permute the sequence used as input or use completely random Gaussian values (keeping the values at each locus the same across cell types), effectively removing any real biological signal, we see the same trends (Additional file 2: Figure S2). Third, comparison to a suitable baseline predictor—namely, the average expression value associated with a given locus across all cell types in the training set—outperforms any of the trained models (solid yellow line in Fig. 1e). Thus, it seems that the more complex neural networks achieve good performance by effectively remembering which genes tend to exhibit high or low expression across cell types. Furthermore, though we demonstrate here that models may use nucleotide sequence to memorize gene activity, the phenomenon is more general, in the sense that any signal that is constant across cell types can be exploited in this fashion. Examples include features derived from the nucleotide sequence—$k$-mer counts, GC content, nucleotide motifs occurrences, or conservation scores—or even epigenomic data when the input is signal from a constant set of many cell types rather than a single cell type.

It is worth pointing out that, from a machine learning perspective, the neural network is not doing anything wrong here. On the contrary, the neural network is simply taking advantage of the fact that most genomic or epigenomic phenomena that are subjected to machine learning prediction exhibit low variance, on average, across cell types. For example, the gene expression level of a particular gene in a particular cell type is much more similar, on average, to the level of that same gene in a different cell type than it is to the level of some other gene in the same cell type. Similarly, many transcription factors bind to similar sets of sites across cell types, most pairs of promoters and enhancers will never interact, and most regions of the genome are unlikely to ever serve as TAD boundaries.

This pitfall can be identified in several ways. First, comparison of model performance to an appropriate baseline, such as the average activity in the training cell types at the given locus (yellow lines in Fig. 1e, f, h, i), will often show that an apparently good model underperforms this relatively simple competitor. As an example, this average activity baseline outperforms two of the top four participants in the ENCODE-DREAM transcription

factor binding challenge (https://www.synapse.org/#!Synapse:syn6131484/wiki/402026) at predicting CTCF in the iPSC cell line when the models were evaluated on loci that they were also trained on (Additional file 2: Figure S3). Notably, CTCF is an outlier among DNA-binding proteins due to its strong specificity for binding at the CTCF motif and similar binding patterns across most cell types. If the trained machine learning model cannot outperform this "average activity" baseline, then the predictions from this model may not be practically useful.

Second, the performance of the model can be more fully characterized by partitioning genomic loci into groups according to their variability across cell types and then evaluating model performance separately for each group (Additional file 2: Figure S4). This partitioning removes the predictive power of the average activity; thus, models that have memorized this average activity will no longer perform well. Indeed, we observe that models that use only nucleotide sequence appear to perform well in the cross-cell type setting but perform markedly worse when evaluated in this partitioned manner.

Several approaches may improve the cross-cell type predictive performance of models that underperform the average activity baseline. A natural approach is to use the average activity directly when training a machine learning model, as Nair et al. [18] do. Another approach would be to phrase the prediction problem not as predicting the activity directly, but predicting the difference from the average activity at that locus for that specific cell type. This approach allows the model to focus on learning cell type-specific differences.

Although most cross-cell type predictive tasks would benefit from a comparison to the average activity baseline, it is important to note in some settings beating the average activity baseline is not necessary. One such setting is the semi-supervised setting, where only a portion of labels are known in advance and the goal is to identify previously unidentified annotations. In this case, because the full set of true labels is not known in advance, a comparison to the average activity may be a poor estimator of the ability of the model to identify novel elements. A second setting is that of anomaly detection, where one identifies regions that are poorly modeled for further study. In each of these settings, it is still informative to compare the performance of the models to the average activity baseline to demonstrate the strength of the predictive model.

Naturally, the strength of the average activity baseline will depend on the degree of similarity between the cell types in the training and test sets (Additional file 2: Figure S5). Hence, it is important for both the developers and users of models to explicitly consider the cell types used to train the model and their anticipated similarity to the cell types that the model will be applied to. For example, a model that is trained using immune cells may exhibit good performance when applied to other immune cells; however, if the model relies too heavily on learning the average activity (a very useful signal in this case), it will fail to generalize to non-immune cells. Conversely, even cell types that are functionally distinct from one another may have some forms of biochemical activity that are surprisingly similar. Even CD8 naive primary cells, which have the most dissimilar gene expression pattern to H1 of the cell types we considered, still achieve an average precision of 0.818 when predicting H1 gene expression.

As more data becomes available, we anticipate that more projects will risk suffering from the pitfall that we describe. Fortunately, avoiding this trap is straightforward: compare model performance to a baseline method that extracts the experimental signal from one or more training cell types, as has been done by several studies working on

cross cell-type prediction [17, 19, 20, 23]. As we have argued here, this comparison is a necessary component of demonstrating the utility of the model.

## Supplementary Information

---

**Additional file 1:** Methods. Details related to the datasets used and the training and evaluation of the machine learning models presented in this work.

**Additional file 2:** The supplementary figures and tables references in this work.

**Additional file 3:** The review history.

---

**Acknowledgements**

We would like to thank Anshul Kundaje and Akshay Balsubramani for providing data from the ENCODE-DREAM challenge.

**Peer review information**

**Review history**

The review history is available as Additional File 3.

**Authors' contributions**

WN, JS, and RS designed the experiments. JS performed the experiments. WN and JS wrote the manuscript. All authors read and approved the final manuscript.

**Authors' information**

Twitter handles: @thabangh (William Noble); @jmschreiber91 (Jacob Schreiber).

**Funding**

This work was funded by National Institutes of Health awards U24 HG009446 and U01 HG009395.

**Availability of data and materials**

The ChIP-seq and RNA-seq signal values were downloaded from the Roadmap compendium (https://egg2.wustl.edu/roadmap/data/byFileType/signal/consolidated/macs2signal/pval/ and https://egg2.wustl.edu/roadmap/data/byDataType/rna/expression/ respectively). The GENCODE gene annotations can be found at https://www.gencodegenes.org/human/release_19.html. Accession numbers associated with supplementary figures are included in the Additional Files.

**Ethics approval and consent to participate**

Not applicable.

**Consent for publication**

Not applicable.

**Competing interests**

The authors declare that they have no competing interests.

**Author details**

[1]Paul G. Allen School of Computer Science & Engineering, University of Washington, Seattle, USA. [2]Department of Genome Science, University of Washington, Seattle, USA. [3]Current Affiliation: Department of Computer Science, and Center for Computational Molecular Biology, Brown University, Providence 02906, RI, United States. [4]Department of Electrical & Computer Engineering, University of Washington, Seattle, USA.

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

## 

