## [**Additional file 3** The review history. · Genome Biology]

Review history

First round of review

Reviewer 1

In this Short Report, Schreiber et al. discuss an important observation about measuring the performance of complex machine learning models in their ability to predict functional annotations from sequence features. Namely, models that measure performance "across cell types" tend to memorize average activity of a given genomic position, and potentially lack the ability to truly generalize. They discuss a hybrid evaluation approach (cross-chr & cross-cell-type) that alleviates the observed issue.

It's unfortunate that this observation has not been noted earlier. Given that several prominent papers, as cited by the authors, have failed to investigate this, I think this short report bares an important message to the community and in theory Genome Biology is an appropriate venue for publication. However, I have some major comments/questions about the presentation and evaluation of the results.

- In the cross-cell-type analysis, one important "parameter" that the authors don't mention or investigate is the similarity between "test" and "training" cell types. It's obvious that if test cell type is completely different from any of the training cell types, memorizing results are not going to be helpful (e.g., "expression profiles" from primary brain and blood cell types have very little correlation). I am also concerned about the high performance of "Average Activity Baseline" in figure 1. This means that the cell types included are overall very similar.

- Related to above, and at a higher level, the overall goal of predicting "signal" for un-seen cell type without a quantification of similarity between training and test cell types seems highly flawed --- i.e., it is hard to interpret any result without knowing the similarity between test and training cell types: so any metric that quantifies performance should also have a notion of cell type similarity.

- It wasn't clear why "random baseline" in Figure 1d and 1e differs? Can you describe what exactly "random baseline" approach is? Another relevant baseline could be an equivalent model but trained on shuffled sequences (i.e., "random" input), or perhaps with shuffled training data to break the link between region and average expression.

- In evaluation the "generalizability" of a model, one can consider two separate issues: 1) is the model's performance, e.g., measured by average precision, over-estimated? 2) are the features learned by the model meaningful? The authors are only concerned with (1) here, but in most biological applications the main utility is (2). As authors know, a model that has potentially over-fitted can still have learned useful biological features. Would be relevant to ask if features learned by the "cross-cell-type" model is vastly different from those of hybrid model.

Reviewer 2

The authors present a key pitfall of performance evaluation of machine learning methods developed for predicting genome-wide measurements across biosamples (cell types/conditions etc.). They show that the a trivial genome-wide summary statistic (such as an average) derived from the genomic measurements on training samples is often a strong baseline. Many published methods in the literature do not compare to this strong baseline and often do not outperform it. The authors provide several useful case studies of the importance of comparing to this baseline. While this baseline may appear 'obvious' to some or trivial to other

in hindsight, I think it is important to explicitly publish this short exposition as it will help elevate the issue and allow the community to refer to it as a standard for performance evaluation.

Authors' response to reviewers

Reviewer 1

In this Short Report, Schreiber et al. discuss an important observation about measuring the performance of complex machine learning models in their ability to predict functional annotations from sequence features. Namely, models that measure performance "across cell types" tend to memorize average activity of a given genomic position, and potentially lack the ability to truly generalize. They discuss a hybrid evaluation approach (cross-chr & cross-cell-type) that alleviates the observed issue.

It's unfortunate that this observation has not been noted earlier. Given that several prominent papers, as cited by the authors, have failed to investigate this, I think this short report bares an important message to the community and in theory Genome Biology is an appropriate venue for publication. However, I have some major comments/questions about the presentation and evaluation of the results.

- In the cross-cell-type analysis, one important "parameter" that the authors don't mention or investigate is the similarity between "test" and "training" cell types. It's obvious that if test cell type is completely different from any of the training cell types, memorizing results are not going to be helpful (e.g. "expression profiles" from primary brain and blood cell types have very little correlation). I am also concerned about the high performance of "Average Activity Baseline" in figure 1. This means that the cell types included are overall very similar.

We agree that the strength of the average activity baseline is dependant on the similarity between training and test cell types. In response to this critique, we now include an analysis of the average activity when recalculated using either the most dissimilar or most similar cell types. Please see our additional text and Figure 1 below (Supplementary Figure 5 in the paper).

Naturally, the strength of the average activity baseline will depend on the degree of similarity between the cell types in the training and test sets (Supplementary Figure S5). Hence, it is important for both the developers and users of models to explicitly consider the cell types used to train the model and their anticipated similarity to the cell types that the model will be applied to. For example, a model that is trained using immune cells may exhibit good performance when applied to other immune cells; however, if the model relies too heavily on learning the average activity (a very useful signal in this case), it will fail to generalize to non-immune cells. Conversely, even cell types that are functionally distinct from one another may have some forms of biochemical activity that are surprisingly similar. Even CD8 naive primary cells, which have the most dissimilar gene expression pattern to H1 of the cell types we considered, still achieve an average precision of 0.818 when predicting H1 gene expression.

- Related to above, and at a higher level, the overall goal of predicting "signal" for un-seen cell type without a quantification of similarity between training and test cell types seems highly flawed — i.e. it is hard to interpret any result without knowing the similarity between test and training cell types: so any metric that quantifies performance should also have a notion of cell type similarity.

We agree that more careful consideration of the similarities between training and testing cell types should be used when training and adopting models. Please see our additional text below.

Naturally, the strength of the average activity baseline will depend on the degree of similarity between the cell types in the training and test sets (Supplementary Figure S5). Hence, it is important for both the developers and users of models to explicitly consider the cell types used to train the model and their anticipated similarity to the cell types that the model will be applied to. For example, a model that is trained using immune cells may exhibit good performance when applied to other immune cells; however, if the model relies too heavily on learning the average activity (a very useful signal in this case), it will fail to generalize to non-immune cells. Conversely, even cell types that are functionally distinct from one another may have some forms of biochemical activity that are surprisingly similar. Even CD8 naive primary cells, which have the most dissimilar gene expression pattern to H1 of the cell types we considered, still achieve an average precision of 0.818 when predicting H1 gene expression.

Figure 1: The performance of the average activity baseline as a function of cell type similarity. The strength of the average activity baseline in the cross-cell type setting at predicting H1 gene expression when calculated using three different approaches. The first approach is to calculate it, not as an average over several cell types, but individually from each cell type in the training set (blue dots) where the average precision is plotted as the y-axis and the correlation with gene expression in H1 is the x-axis. The second approach is to average gene expression values across all cell types that are at least as similar as a certain cell type, i.e. average over all cell types to the right (orange line). The third approach is to average gene expression values over all cell types that are no more similar than a certain cell type, i.e. average over all cell types to the left (magenta).

- It wasn't clear why "random baseline" in Figure 1d and 1e differs? Can you describe what exactly "random baseline" approach is? Another relevant baseline could be an equivalent model but trained on shuffled sequences (i.e. "random" input), or perhaps with shuffled training data to break the link between region and average expression.

Thank you for asking for clarification on this point. The difference in the "random baseline" performance is because chr1 is used as the test set for the cross-chromosomal and hybrid test settings and chr2-22 are used as the test set for the cross-cell type evaluation. By construction, the training and test sets cannot both be the same for the cross-cell type and hybrid (or cross-cell type and cross-chromosomal) settings, and we describe our evaluation procedure in the "Model training" section of the supplement: "We chose to hold the training set constant between the cross-cell type and hybrid approaches, rather than the test set, in order to demonstrate that models trained on the same data exhibit markedly different trends with respect to model complexity depending on the evaluation set." We have clarified this point in the main text.

Note that the random baseline, which is the expected average precision when the predictions are uniformly random values and represents a lower bound of performance, differs between the cross-cell type and other settings because we use a different chromosome for the test set (see Supplementary Note 1 for details).

We agree that training models with shuffled sequences would be valuable. Accordingly, we have added two more evaluations where we train neural networks on sequence data that has been permuted (consistent across cell types) and where the sequence has been replaced with Gaussian random values (also consistently across cell types.) We observed the same trends in both cases because the models are capable of using any input values that are consistent across cell types to memorize the signal. See our new text and Figure 2 below (Supplementary Figure 2 in the paper).

Second, even when we permute the sequence used as input or use completely random Gaussian values (keeping the values at each locus the same across cell types), effectively removing any real biological signal, we see the same trends (Supplementary Figure 2).

Figure 2: The performance of neural networks with randomized inputs. The figure plots the average precision (AP) of neural network models predicting gene expression using randomized inputs as a function of model complexity. The inputs are either one-hot encoded nucleotide sequence that has been permuted (in cyan) or Gaussian random values (in purple). In both cases, the representation for each gene is consistent across cell types. Evaluation is performed via (a) cross-chromosome, (b) cross-cell type, and (c) a combination of cross-chromosome and cross-cell type validation. In each panel, each point represents the test set performance of a single trained model.

- In evaluation the "generalizability" of a model, one can consider two separate issues: 1) is the model's performance, e.g. measured by average precision, over estimated? 2) are the features learned by the model meaningful? The authors are only concerned with (1) here, but in most biological applications the main utility is (2). As authors know, a model that has potentially over-fitted can still have learned useful biological features. Would be relevant to ask if features learned by the "cross-cell-type" model is vastly different from those of hybrid model.

We believe that one should be extremely cautious when interpreting a model that is known to be overfit (e.g. because the model is simply memorizing the average activity). As we demonstrate with the experiments with randomized inputs, a model can still achieve good performance when artifacts in the data set can be abused to achieve good performance. Interpreting such a model would be unwise and unlikely to yield anything biologically meaningful.

Reviewer 2

The authors present a key pitfall of performance evaluation of machine learning methods developed for predicting genome-wide measurements across biosamples (cell types/conditions etc.). They show that the a trivial genome-wide summary statistic (such as an average) derived from the genomic measurements on training samples is often a strong baseline. Many published methods in the literature do not compare to this strong baseline and often do not outperform it. The authors provide several useful case studies of the importance of comparing to this baseline. While this baseline may appear 'obvious' to some or trivial to other in hindsight, I think it is important to explicitly publish this short exposition as it will help elevate the issue and allow the community to refer to it as a standard for performance evaluation.

Thank you for your positive assessment of our work.

Second round of review

Reviewer 1

Thank you for your thoughtful response, my concerns have been addressed. On my last comment "interpreting" features of a model that has potentially overfitted: in your context, depending on the level of overfitting, yes make sense that probably overfitted model's features are irrelevant. My comment was about a more general view, e.g., see discussion of 'Reply to 'Inflated performance measures in enhancer-promoter interaction-prediction methods' --- slightly overfitted models can still find biological signal that may be of relevance.